# Fear, Stigma and Othering: The Impact of COVID-19 Rumours on Returnee Migrants and Muslim Populations of Nepal

**DOI:** 10.3390/ijerph19158986

**Published:** 2022-07-23

**Authors:** Pramod R. Regmi, Shovita Dhakal Adhikari, Nirmal Aryal, Sharada P. Wasti, Edwin van Teijlingen

**Affiliations:** 1Faculty of Health and Social Sciences, Bournemouth University, Bournemouth BH8 8GP, UK; sdhakaladhikari@bournemouth.ac.uk (S.D.A.); naryal@bournemouth.ac.uk (N.A.); evteijlingen@bournemouth.ac.uk (E.v.T.); 2School of Human and Health Sciences, University of Huddersfield, Huddersfield HD1 3DH, UK; s.p.wasti@hud.ac.uk

**Keywords:** COVID-19, migrants, Muslim, moral panic, stigma, fear, Nepal

## Abstract

The paper explores how COVID-19-related moral panics have led to fear and othering practices among returnee Nepalese migrants from India and Muslims living in Nepal. This qualitative study included in-depth interviews with 15 returnee migrants, 15 Muslims from Kapilvastu and Banke districts of Nepal, and eight interviews with media and health professionals, and representatives from migration organisations. Four themes emerged from our data analysis: (1) rumours and mis/disinformation; (2) impact of rumours on marginalised groups (with three sub-themes: (i) perceived fear; (ii) othering practices; (iii) health and social impact); (3) resistance; and (4) institutional response against rumours. Findings suggest that rumours and misinformation were fuelled by various media platforms, especially social media (e.g., Facebook, YouTube) during the initial months of the lockdown. This created a moral panic which led to returnee migrants and Muslim populations experiencing fear and social isolation. Resistance and effective institutional responses to dispel rumours were limited. A key contribution of the paper is to highlight the lived experiences of COVID-19 related rumours on marginalised groups. The paper argues that there is a need for clear government action using health promotion messages to tackle rumours (health-related or otherwise), mis/disinformation and mitigating the consequences (hatred and tensions) at the community level.

## 1. Introduction

The novel coronavirus disease (COVID-19) affected many people’s lives—not only in terms of health, but also economically and socially. Current global health systems are overwhelmed, with people failing to access health services in many countries including Nepal. In Nepal, the cumulative total cases have reached 814,289,914 with 11,438 deaths by 5 November 2021, jeopardising the Nepalese health system [1]. The COVID-19 pandemic has highlighted so-called ‘fake news’ and its ramifications for many disadvantaged and marginalised groups. Rumours and social stigma related to the COVID-19 pandemic tend to increase globally, and a growing number of studies document its repercussions including on people seeking services and support [2]

Misinformation (inadvertently) and disinformation (advertently) are not novel threats to public health, especially during disease outbreaks or natural disasters [3]. There is strong evidence that misleading information has the tendency to spread faster than accurate information through social media outlets [4]. The current pandemic of COVID-19 has a vigorous psychological effect.

Drawing on Cohen’s (1972) notion of ‘moral panic’ in health crises, it is often the case that mis/disinformation fuels society and accurate information is neither timely nor well disseminated to relevant communities [5,6]. According to Cohen (1972) moral panics are ‘boundary crises’ that occur at times of uncertainty [5]. Boundary crises refers to the uncertainty as to where the boundary lies between acceptable and unacceptable behaviour at a times of social change. The length of moral panics differs from case to case; they can be quickly forgotten or have long-term repercussions for society [6]. Cohen (1972) argues that the way in which the situation is initially interpreted and presented by traditional mass media affects how misleading information is exaggerated on other media platforms (e.g., YouTube, Facebook, Twitter), for example during earlier Ebola and Zika outbreaks.

Although critics question the use of the concept of moral panic in understanding social problems in the past few decades, nevertheless it still holds value in exploring media and societal reactions to virus outbreaks [7,8,9].

Information on social media has been shown to reinforce health-related fears and phobias. For instance, there is evidence that at least one-quarter of popular social media content (in terms of shares, likes, visits) is misleading in many health-related messages [10,11]. Many unknowns around COVID-19 and the fear of being infected gave rise to many rumours, which are in itself illness narratives [12], fueling moral panics in local communities.

Mass hysteria developed worldwide which fuelled discrimination and attacks against vulnerable people [13]. Such moral panic was also found in the early years of AIDS (acquired immuno-deficiency syndrome) in the United States of America [14], and Ebola and SARS globally [8]. Ungar (2016; p. 349) refers to moral panics during health crises as “viral moral panics” which are often localized, and “involve efforts to use morality to regulate public behaviours” [8], for example, moral obligations include to stay at home, follow quarantine rules and to be vaccinated. Migrants and other marginalised groups have historically been targets of blame, scapegoating and stigmatisation during disease outbreaks; they are labelled “disease breeders”, which contributes to racism and xenophobia [15,16]. Thus, moral panics create a favourable relationship between the press and the state whereby it enables the state to establish control over a population by leveraging the perceived threat generated by the moral panic [17]. Ungar (2016, p. 349) reminds us that “expressed fears and hostilities are out of proportion to the actual threat” [8].

Such moral panics being based on stereotypes and used to reinforce them by labelling a specific group can be useful in explaining ‘othering’ practices of those who are a threat in the crisis period [5,6,18,19]. Othering is a process that identifies those that are thought to be different from the mainstream, which reinforces positions of domination and subordination [20]. Consequently, persons who are treated as ‘other’ often experience marginalisation, decreased opportunities, and exclusion [20].

Nepal offers an interesting research setting. Ever since COVID-19 became a major issue in South Asia, mainstream mass media in Nepal frequently reported on the risk of COVID-19 being brought to Nepal by thousands of migrant workers returning from India, Muslims from India travelling to Nepal for religious activities, and Nepalese Muslims attending religious gathering in India [21]. Muslims have a history of being discriminated against on the pretext of their involved in terrorism [22,23]. It was further propagated by social media enforcing moral panic and promoting negative public perception against these two marginalised communities as possible virus carriers and spreaders. For instance, Sijapati (2020) stated that Nepalese migrants (especially in India) and Muslims living in the border areas with India were to blame for bringing COVID-19 to Nepal [21]. Studies have shown how mass media and social media’s presentation/interpretation of disasters/crises promote both moral panics and public fear [19,24,25]. Moral panics often target marginalised groups based on ethnicity, class, or religion [5,19,26]. Exploring how moral panics are created around already marginalised and vulnerable populations can help explain ‘othering’ practices [5,6,19]. The impact of COVID related rumours is yet to be empirically supported, particularly the lived experiences of the marginalised groups, these constitute knowledge gaps which this study intends to address.

This is the first study of its kind to explore: (a) how moral panics related to COVID-19 have led to rumour-induced fear, and othering practices among returnee Nepalese migrants and Muslims in Nepal; and (b) its impact on their overall wellbeing and their coping/managing and resistance strategies against othering practices.

## 2. Methods and Materials

### 2.1. Study Design and Participants

To explore moral panics related to COVID-19 and its consequences, an exploratory qualitative study using interviews was designed. In early 2021, we carried out interviews with 15 returnee migrants and 15 Muslim individuals. These participants were based in two districts (Kapilvastu and Banke) in the Terai (southern Nepal), both bordering India, and both with a higher-than-average proportion of migrants and Muslims. In addition, eight interviews with key stakeholders were carried out. Our study interview participants were: (a) migrants from India, who had returned in the six months prior to the interview, all aged 18 years and above; (b) Muslim residents in the districts aged 18 years and above; and (c) key stakeholders such as journalists reporting on migrants or Muslim issues during COVID-19, health workers, and representatives working with migrants and marginalised populations in Nepal. All participants were explicitly selected to include diverse backgrounds based on gender, education, employment and age.

### 2.2. Study Tools and Interviews

Based on the literature, the research team—in collaboration with key stakeholders in Nepal-drafted an interview guide in a question route [27] to facilitate the interviews. The interview guide was pre-tested [28] before going live with the study. Our interview schedules included questions on demographics, discrimination, rumours and its sources, coping strategies, and societal preventive measures. All three interview schedules are available on request from the first author. All interviews were conducted in Nepali by experienced and same-sex qualitative researchers [29]. Interviews were carried out in a mutually agreed safe place.

### 2.3. Data Organisation, Management and Analysis

We audio-recorded all of the interviews with permission from the participants. Interview recordings were transcribed and translated into English. Transcripts were cross-checked with original recordings by authors: PR, NA, SW, and SDA. Any disagreements were discussed in detail between team members for appropriate translation. Each transcript had a cover note describing the interviews, settings, the nature of the discussion, any differences from other interviews, particular incidents, the interview environment, and a reflection on the issues identified in the session. We performed thematic analysis [30] and relevant quotes are provided to support the themes. Three groups of interviewees are in the quote identifier with a number (e.g., P3 Muslim).

### 2.4. Ethical Consideration

We received ethical approval from both Bournemouth University, UK and the Nepal Health Research Council (NHRC), Nepal. Through a participant information leaflet (PIL) in Nepali, participants were provided with enough information about the study purpose and procedure, voluntary participation, confidentiality, risk and benefits to the participants, and a complaint procedure [31].

### 2.5. Characteristics of Interview Participants

Table 1 shows that participating migrant returnees and Muslims were mostly male (83.3%) and relatively young (average age 31.1 years, range 18–65 years). A higher proportion of participants were Muslims (56.7%). Just seven participants (23.3%) had completed a higher secondary level of education or above, and the majority (60%) lived in India as migrant workers, predominantly worked in the hospitality sector. Similarly, of the eight stakeholder participants, seven were males; two were journalist, one representative of an international non-governmental organisation (INGO), three represented a non-governmental organisations (NGO), and two represented government organisations (GOs) working in the field of migration. These stakeholders had an average of 11.6 years of work experience. 

## 3. Key Findings

Our analysis revealed how COVID-19 rumours have impacted our participants’ wellbeing and their relationships with the community. We present the findings in four sections. First, we examine the various rumours and mis/disinformation propagated by media outlets which created the moral panic. We then present the impact of these rumours on marginalised groups including- perceived fear, othering practices-stigma, discrimination, abuse, humiliation, and blame, and their implications on health and social wellbeing. We then examine the ways our participants responded to these rumours and othering practices. Finally, we consider the institutional responses to tackling the rumours.

### 3.1. Rumours and Mis/Disinformation during the COVID-19 Pandemic

Rumours and misinformation were fuelled by social media and online news portals during the initial months (April-June 2020) of the lockdown in Nepal. Most of the participants blamed Indian television channels, Nepali online news portals, and social media (mainly Facebook and YouTube) for spreading rumours portraying Muslims as carriers and propagators of the COVID-19. For example, a Muslim participant said:


*I think Indian media, YouTube videos, and Facebook posts were the main source of misinformation. Many of the local people whom I knew shared social media pages and videos blaming Muslims for corona spread, Muslims being beaten, and so on. Those sources of misinformation were mostly from India.*
[P 1-Muslim]

Another participant argued that some Nepali online news portals were also responsible for anti-Muslim rumours:


*Indian media and some Nepali online news portals were responsible for anti-Muslim rumours. I still remember the name of these media; I can name them but am not aware of any Nepali television, radio channels, and newspapers that provided misinformation against the Muslim community.*
[P 2-Muslim]

Key stakeholder and journalist participants supported the views of migrants and Muslims. They believed that Nepali mainstream media could not effectively tackle rumours and mis/disinformation, and that there were no systematic efforts by government organisations and NGOs to tackle media rumours in Nepal. A representative of an NGO working for migrants said:


*I think social media, primarily Facebook, played a vital role in spreading rumours and mis/disinformation against migrants and Muslims. In recent years, development and the role of media in society has been very commendable. I think mainstream media should have been more proactive to dispel any rumours during the pandemic.*
[P 35-Key stakeholder]

In response to a question to explore the role of NGOs, participants noted that NGOs did not contribute to awareness raising around COVID-19 during the initial period in 2020, nor helped to dispel myths and rumours. As quoted by a public health officer:


*NGOs were passive for the initial three to four months. They could have provided support on raising awareness, use of personal protective equipment (PPE), and other support. I think schoolteachers could have been mobilised in quarantine shelters, for awareness or where support was needed. However, this did not happen.*
[P 33-Public Health Officer]

Consequently, these rumours and mis/disinformation propagated by various media outlets enforced moral panic-promoting negative public perception against these two marginalised communities as possible virus carriers and spreaders.

### 3.2. The Impact of Rumours and Mis/Disinformation on Marginalised Groups

Our analysis revealed how COVID-19 rumours have impacted upon our participants wellbeing and their relationships with community in several forms: perceived fear, othering practices such as stigma, discrimination, abuse, humiliation, and blame, and implications on health and social wellbeing, such as quarantine, hearsay information, and emotional impact on individuals.

(i)Perceived Fear

Due to the fear of being victimised by the community, some Muslims participants felt that people acted differently than usual. Migrants also stated that, despite having negative test results for COVID-19, they usually conducted a self-imposed home quarantine in addition to the mandatory two-weeks stay at a government quarantine facility.


*Local villagers said that I brought Corona from India and warned me not to come near. They did not show human behaviour to me. I did not come outside the home for 14 days. I did not ever feel so hurt (pause…), earlier they used to come to my house to see me when I returned back.*
[P 2-Migrant]

In the later stage of COVID-19, the local government allowed returnee migrants to opt for home quarantine as an alternative to self-isolation at the government centre. However, due to potential hostility from the local villagers, some still opted to stay at government quarantine, for example:


*Although our local government allowed returnee migrants to stay in home quarantine, I stayed in government quarantine first because villagers could accuse me of transmitting COVID-19 later on. Even after I returned to the village, I did not go outside of the house for five days.*
[P 3-Migrant]

Muslim participants generally worried about what people would say or act against them or label them as COVID carriers much to the detriment of their mental health and wellbeing:


*I lived with fear after my return from India. I was scared that someone would call the police and lodge a complaint against me. I was mentally disturbed during that period.*
[P 1-Muslim]

Muslims feared rumours may have an influence on people who were otherwise close to them:


*I was always in fear if someone would say anything to me or abuse me while travelling. I also wondered if these rumours would change the attitude of non-Muslims towards us, who otherwise were very close and connected.*
[P 14-Muslim]

A key informant participant provided a detailed example:


*The first case of COVID-19 in Banke district was diagnosed in a 60 year old Maulana (Muslim religious leader) who used to teach Urdu in the community and had just returned back from India. A further five, six cases were identified from his contact tracing. This had created a chaotic situation among Muslims and people blamed them for bringing corona in Banke. Three major Muslim dominant suburbs were cordoned off. This mainly happened during the first phase of corona.*
[P 31-Journalist]

(ii)Othering practices: Stigma, discrimination, abuse, and blame

Our findings reveal that returnee Nepalese migrants from India and Muslims were frequently labelled as COVID carriers. Migrant participants shared that they had experienced various forms of discriminatory practices from the villagers in transit, during their quarantine stay, and from the local community after their return. Some of these behaviours barred the migrants from purchasing basic commodities such as food or mobile phone vouchers. Below are two quotes from our migrant participants:


*While returning to Nepal we were told not to touch the water tap. Also, shopkeepers refused to sell us any food because they thought that the banknotes which we were carrying might have corona. Despite having money, we could not buy any food.*
[P 22-Migrant]


*We could not buy anything from outside while staying in quarantine. Shopkeepers refused to take currency notes from us as they believed that these notes might be infected with corona. We could not buy even a recharge card for our mobiles.*
[P 28-Migrant]

Muslim participants also shared stigmatising and discriminatory behaviours in their community.


*Muslims are easily identified by their looks and appearance. I had experienced malignant behaviours from both whom I knew and people new to me. People who were usually friendly to me started discriminatory behaviour. In teashops and restaurants, they told me not to come near them although they were sitting closely in a group. I felt that my customers also talked to me from an unusually far distance when they knew I was a Muslim.*
[P 1-Muslim, Shopkeeper]

There was a striking example of discriminatory behaviour to Muslims from the local government:


*A Muslim religious preacher came from Pakistan four months before the lockdown and was living in a local Madrasa (Muslim educational institution). Nonetheless, the local administration forced him to stay in quarantine for 14 days due to pressures from the local community.*
[P 2-Muslim]

In concurrence with the views of both migrants and Muslim participants, our stakeholder participants also reported stigma and discrimination against these populations, stating:


*Returnee migrants have said that they did not have to face discriminatory and stigmatising behaviour in India, however, have experience such behaviour in Nepal after return. There were a few incidents in which even family members did not accept them. Nothing could be as unfortunate as this for returnee migrants.*
[P 35-NGO Representative]

Most of the migrants repeatedly stated that discrimination and blaming within the migrant population varied. Strikingly, migrants who were poor, Dalit, and female were affected more than others. An NGO worker mentioned:


*Migration to India is very common among Dalit (oppressed) populations as most of them have no agricultural land. I feel they experienced the double whammy of rumours/stigma as a returnee migrant and also as a Dalit.*
[P 36-NGO Representative]

On the other hand, both migrants and stakeholders expressed that the community behaved differently towards returnee migrants from India than those from other countries (e.g., USA, Gulf countries) and that the latter groups did not face much discrimination.


*Society did not discriminate against returnee migrants from the USA, Italy, the Gulf countries and Malaysia as they did for returnee migrants from India. During the initial phase of the COVID pandemic, returnee migrants from India were not allowed to enter their own house in Banke [=district name] even after spending 14 days in quarantine. Some of them stayed on the open roof of their house.*
[P 31-Journalist]

Due to fear of being victimised or stigmatised in transit, some migrants entered Nepal using alternative routes rather than official border checkpoints. For example, a migrant worker elaborated on this:


*When we crossed the border, border-side villagers in Nepal did not allow us to enter Nepal, fearing possible corona transmission from us. We had not eaten anything for four days but had to walk hours to cross the border via side routes (agriculture field) so as to evade villagers.*
[P 27-Migrant]

Key stakeholder participants also supported these incidents:


*We used to bring returnee migrants in the trucks and tractors from the Indian border to quarantine shelters in Nepal. Local people living around the main roads and side roads warned us not to use the road to carry migrants because they thought it might transmit COVID by just using the road. They even threatened us with possible offensive actions.*
[P 34-Public Health Officer]


*In a number of places, local people erected a barricade on the road to restrict the return of migrants. People believed that vehicles carrying migrants would transmit COVID-19. Personally, I also tried to convince them not to close the roads.*
[P 32-Journalist]

After arriving in Nepal, not only community members but also locally elected leaders and local government officials treated the migrants differently. Both internal migrants and those from India experienced roadblocks preventing them from going home since they were perceived as COVID-19 carriers. For example, a journalist stated:


*There were issues with internal migrants as well. People working in different parts of the country walked for several days to return back home because public transportation was not available. When they arrived at the border of Banke, related municipalities did not allow them to enter, fearing COVID transmission. At first, they were dropped by Kohalpur municipality to the border of another municipality using a tipper truck. At least four, five municipalities did the same consecutively and eventually they were returned to the Kohalpur municipality which had first done that. None of the municipalities allowed them to enter.*
[P 31-Journalist]

Interestingly, these behaviours gradually decreased after a rapid surge of COVID cases in areas with a negligible number of migrants.

(iii)Implications on health and social wellbeing

We found that both migrants and Muslims did not generally experience problems while accessing health care services. A few Muslim participants however refrained from accessing health services due to the fear that they would be forced to stay in quarantine. They preferred home quarantine over publicly-run quarantine centres:


*I do not think there were any issues with accessing health care, however, during the initial phase [of COVID], they did not go to the hospital even when they were seriously ill, due to the fear of being kept in quarantine. Instead, many of them contacted Muslim health care workers for medication.*
[P 15-Muslim]

Almost all participants alluded that they did not experience any discriminatory behaviour from health care providers while accessing health services, with some exceptions. For example, a Muslim participant expressed that some service providers treated Muslim patients differently when providing services:


*I have heard that some Muslim community members were neglected while seeking treatment in local health centres. However, I have not experienced any such incident.*
[P 2-Muslim]

There were mixed responses from the migrant participants about their experiences of staying in quarantine centres. For example, some were positive:


*I stayed in Chandrauta quarantine [Kapilvastu district], tested positive and lived in isolation for 14 days where the facilities at quarantine were good, with no issues at all. Health workers also visited from time to time.*
[P 20-Migrant]

Whilst other migrants had negative experiences:


*Not a single health professional turned up to see us in quarantine. I felt abandoned and discriminated against for being a returnee migrant.*
[P 21-Migrant]

Generally, migrants felt comfortable re-integrating into society after quarantine and homestay, although there were a few exceptions to this:


*Some villagers used to tell me to stay away from them as I was back from Bombay [=Mumbai] and they would get corona if I touched them.*
[P 19-Migrant]

Furthermore, in response to discriminatory practices, some Muslims indicated that they felt a loss of “dignity” and “confidence” and were “emotionally hurt” due to social blaming. A Muslim participant stated:


*One Muslim member with a hotel business died during the second lockdown. It was not clear whether he died from COVID-19 or not. His body was not handed over to the family for the last ritual according to the Muslim religion, instead was buried somewhere else. We were ready not to open the plastic-wrapped body and follow every safety measure in the presence of police but were not heeded upon. This incident really hurt me (speaker visibly upset), our entire community was hurt.*
[P 2-Muslim]

Similarly, one Muslim participant stated that COVID-19 had also fuelled community tensions:


*COVID-19 related circumstances have fuelled community tensions between non-Muslims and Muslims and some sort of hatred between them. This should not happen and any activities to incite this should be taken control of on time.*
[P 1-Muslim]

Key stakeholders also agreed that there was emotional impact on Muslims due COVID-19 rumours. For example:


*The Muslim community did not retaliate when it came to rumours against them. However, I found that they were emotionally hurt. No communal tensions had emerged.*
[P 31-Journalist]

### 3.3. Resistance

Most interviewees provided eyewitness accounts on the resistance methods of returnee migrants from India. Strategies for coping with and managing rumours took various forms. To minimise the potential of being othered, some migrants showed their COVID-19 test results to people in their community to confirm that they did not have the virus. Muslims attempted to resist by talking to local community leaders, and some were also involved in counterarguments on social media. The latter was suggested in this example:


*I have talked with non-Muslim leaders, friends and intellectuals about the COVID-19 related rumours against Muslims and requested to convey appropriate information about it in the community.*
[P 2-Muslim]

Although rare, some resistance against Muslim- related rumours, i.e., *Muslims would carry COVID-19* was also shown by the local administration. In contrast, a Muslim participant ended up blaming migrants for spreading COVID-19:


*One fellow female passenger in an auto-rickshaw asked me to stay away, blaming that Muslims would carry corona. In another incident, I was standing in a queue in the bank. A security guard said to me that Muslims were responsible for the propagation and spread of corona. On both occasions, I have argued that Pahadiya [People from hilly region] were responsible because they go to India for work and bring corona to Nepal.*
[P 13-Muslim]

However, no formal complaints were lodged by Muslims and migrants against hate speech and derogatory remarks against them, as shared by two key informants:


*I have not heard that Muslims/migrants lodged a formal complaint against discrimination and hateful behaviour.*
[P 32-Journalist]


*I have not noticed any cases that returnee migrants lodged a formal complaint against discriminatory behaviour.*
[P 36-NGO Representative]

### 3.4. Institutional Response against Rumour and Mis/Disinformation

We found that, during the initial months of COVID-19 in Nepal, the institutional response was slow, this includes the Ministry of Health and Population, the various health NGOs and traditional mass media. Ordinary people in the community were often opposing the quarantining of returnee migrants and Muslims and wanted the local government to pay attention to them. Worryingly, these authorities and local politicians also blamed migrants and Muslims for spreading COVID-19. A Muslim participant stated:


*During the initial days of COVID-19, people in authority had blamed the Muslim community as a spreader of this pandemic. I have seen and heard this personally…as time passed this attitude was gradually changed.*
[P 3-Muslim]

Our participants expressed disappointment that Nepali mainstream media could not effectively tackle rumours and mis/disinformation around COVID-19. For example, a journalist said:


*Perhaps we failed to provide information that COVID-19 is not fatal and like other diseases could be prevented with preventive measures. The general public cannot evaluate the source and veracity of the information. They believe in mis/disinformation in social media.*
[P 31-Journalist]

There was also a clear lack of government initiative to monitor and tackle health-related and other rumours. As one Muslim participant stated:


*Government has cyber law and cyber-crime branches. They should be vigilant on any mis/disinformation against any community. Such media outlets should be punished, and their webpages should be blocked.*
[P 2-Muslim]

Almost all participants emphasised the need to take punitive actions against those spreading rumours. For example, a key informant commented that:


*We need to have effective preparedness and mechanisms to handle rumours and misinformation. Actions were also taken against a few online new portals for spreading rumours during COVID-19. Maybe we have to apply stringent measures against those spreading rumours.*
[P 35-Public Health Officer]

With a few exceptions, participants noted that positive initiations were led by local administration to quash rumours against Muslims. For instance, a public health officer said:


*Local government issued a statement that COVID-19 was not related to any religion, community, group and anyone can contract it. I was really happy seeing this statement.*
[P 34-Public Health Officer]

Participants suggested different preventive measures to dispel dis/misinformation. They further suggested that methods to deal with rumours should be placed at the governmental level. For example:


*I think the source of misinformation is the media, and media could play a role to minimise it as well. For e.g., programmes on radio, television, content in the newspaper, and even public miking could highlight that COVID related issues are religion-related.*
[P 1-Muslim]


*Muslim organisations and I had used social media platforms (Facebook, Twitter) to dispel rumours against Muslims …I think multiple approaches using social media and street drama would be effective to tackle misinformation against Muslims.*
[P 4-Muslim]

A public health officer suggested public loudspeaker announcements to raise local awareness as the literacy rate in some communities is very poor:

*Rautahat district has one of the lowest literacy rates in the country and thus I think that public miking would be more effective to raise awareness here*. [P 34-Public Health Officer]

Participants also expressed that awareness-raising programmes should be led by healthcare professionals whose messages are highly trusted by the community people.


*Health professionals have a key role to tackle misinformation against Muslims because people believe and respect them. They could have given proper information in the community and media that COVID was not related to any religion.*
[P 15-Muslim]

Using a telephone ringtone with COVID-19 message, spreading awareness in the local language via local radio stations, and galvanising local leaders were other frequently suggested strategies by our participants. Some also suggested educating or even galvanising religious leaders, political figures (especially elected representatives at the ward levels) and administrative staff of the community as they were involved in inappropriate actions:


*Local administration could have discussed with community/religious leaders of both Muslims and non-Muslims about any mis/disinformation and misconceptions on COVID-19. People largely follow religious leaders and if they are apprised properly, it could be effective.*
[P 1-Muslim]

## 4. Discussion

Using the notions of moral panic and othering practices this study has investigated how COVID-19 rumours have fuelled panic and fear, resulting in discrimination towards marginalised groups (i.e., returnee Nepalese migrant workers from India and Muslims) [5,7,8,18,20]. This in turn exacerbated vulnerability and social exclusion. In this paper, we argue that rumours and mis/disinformation against marginalised groups have not only impacted their wellbeing but also their relationships with the community and their health-seeking behaviours.

Rumours and misinformation about COVID-19 targeting marginalised population groups is not a new phenomenon [32,33,34]. Migrants and many other disadvantaged groups have historically been targets of blame and scapegoating during disease outbreaks as “disease breeders”, which may further increase racism and xenophobia against these groups [15,16]. For example, a recent study in six low-and middle-income countries (including Nepal and India) to assess impact of lockdown measures during the COVID-19 reported that migrant workers and religious groups were among those who suffered most [35]. However, few studies so far have empirically explored the impact of media induced COVID-19 rumours on marginalised groups. For instance, Arin et al. state that COVID-19 increased misperceptions, and reduced trust towards Muslim communities could be partly attributed to fake news and misinformation [36]. In this regard, our study provides context-specific novel insights into the impact of COVID-related rumours particularly the lived experiences of the marginalised groups.

Our findings demonstrate that not only the general population, but also higher-ranking officials and political leaders blamed Muslims for the spread of COVID-19. We argue that during periods of uncertainty, or in any such epidemic or pandemic scenario, media and responsible authorities should be sensitive to the scapegoating of marginalised groups, as suggested in Cohen’s theory of boundary crises [5].

During any pandemic, people are desperate for information on risks associated with the disease, its severity, and possible preventive and curative measures. Our findings support the argument advanced by scholars showing that the social media platforms, online news portals and TV played a central role in spreading mis/disinformation, rumours about COVID-19, and even hate speech [37,38]. The news played an influential role in transforming Nepalese returnee migrant workers from India and Muslims into the threat of COVID-19 spreaders in Nepal. This increased media attention targeting the marginalised populations as possible COVID carrier could be argued to fuel moral panic [25,39].

It has been observed that stigmatising behaviour against Muslims and returnee migrants spiked during the initial phase of COVID-19. This was also due to the early COVID-19 positive cases in these population groups. The identification of ‘folk devils’ [5] and their stigmatisation is a feature of moral panic. As COVID-19 cases spread among other population groups, negative perceptions towards Muslims and returnee migrants also gradually declined in tandem with the diminishing negativity in mainstream and social media [9].

While previous studies focused on misinformation and fake news during the pandemic [11,25,36], our study extends our knowledge of lived experiences of othering practices due to COVID-19 related rumours against the marginalised groups. Our findings show both migrants and Muslims felt forced to take safeguarding actions (i.e., doing the best they could under the circumstances). Such actions included self-imposed home quarantine, crossing the border through informal routes (which brings other Public Health risks), and flaunting their COVID-negative test in public. Similar issues were also reported among Indian migrants who faced fears and discrimination in their home communities in India [32]. These are referred to as ‘moral obligations’ to regulate people during health crises [8,39].

Blaming Muslims as COVID-19 spreaders and ostracising ‘the other’ can be perceived as a way for people to make sense of mysterious diseases [40]. This finding is in line with the assumptions that epidemics sparked blame of the ‘other’ and that it would be at its worst when diseases were mysterious as to their causes and cures [41]. Blaming Muslims during the initial phase of COVID-19 was also reported in India, which resulted the entire Muslim community stigmatised as the virus spreaders. The stigmatisation of the whole Muslim community had been at the forefront of COVID-19 and an event by the Islamic missionary movement Tablighi Jamaat was labelled a “corona terrorism”, which fuelled the feelings of hatred [42]. These representations have led to increased fear among them, mostly due to their communities’ untoward reactions (e.g., blaming them as COVID carriers).

Our findings provide evidence that negative attitudes towards returnee migrants from India and Muslims leads to increased tensions (including disturbance to communal harmony) and lower quality of life. The COVID-19 pandemic has resulted in psychological problems such as anxiety, depression, and panic in both migrants and Muslim communities as well as health service providers in Nepal during the early days of the pandemic. We found that local communities behaved differently towards returnee migrants from India than migrants from other countries (e.g., USA, Gulf countries) which resulted in many returnee migrants from India to confine themselves within their houses. Fear of being rejected as a ‘virus carrier and spreader’ by their neighbours and the wider community upon their arrival from India was the salient reason for this. This may have impacted their mental health and wellbeing significantly. As suggested by prior work, for example, higher psychological distress was reported among Nepalese doctors since their communities mistreated them as possible COVID-19 virus carriers and spreaders [43,44]. They were verbally abused, experienced discrimination from their landlords, and faced unilateral termination of tenancy agreement(s) [45]. This demonstrates that a lack of clear information agitated fear and panic among the people, thus necessitating a pragmatic delivery of relevant information and education to communities in the event of any future pandemic.

In the light of significant impact of stigmatisation on the health and health-seeking behaviours of marginalised individuals, and decreased dignity and confidence in society in Muslims and migrants alike due to social blaming [2,42], our study serves as an illustrative example of these. We found that Muslims and returnee migrants both suffered, perhaps more so in returnee migrants from India, Dalits and female migrants. This could be since they are among the poorest, least educated, and seldomly heard groups. Muslims, on other hand, often occupy better socioeconomic positions relative to the returnee migrants, and are thus able to confront rumours in person, via social media, and approaching influential people from other communities.

It has been observed that panic escalates due to a lack of information from the experts and a delay in the dissemination of accurate information to the community [6]. Similarly, despite positive public health efforts by the Nepalese government toward preventing and controlling COVID-19, there were no mechanisms to tackle the misinformation generated and propagated by media during the initial stage of the pandemic. In the latter stage of COVID-19, there were some sporadic efforts from the government and media-related organisations to tackle rumours. However, such efforts were neither systematic nor regular.

## 5. Strengths and Limitations of the Study

Our study is the first to examine rumours regarding the effect of COVID-19 on Muslims in Nepal and returnee migrants from India. We have included wide ranging participants from our study groups to capture diverse views. Moreover, same sex researchers facilitated the interviews to make our data rich, as the study topic was somewhat sensitive in the context of Nepal. In this study, we explored the participants’ experiences in the early days of the COVID-19 pandemic. Our interviews were carried out in the two bordering districts of Nepal, and hence these findings may not represent wider Nepalese migrants or Muslims communities. Most of our interviewees are males, as most migrant workers in Nepal are males and Muslim women often face restrictions in their travel. We suggest that further, broader studies are needed to understand the magnitude of the impact of COVID-19-related misinformation, fear, panic or stigma on the mental health, communal harmony and general wellbeing of migrants and Muslim populations.

## 6. Conclusions

Misinformation fuelled by rumours, mis/disinformation, and stigmas have potentially severe implications on public health. In this paper we have shown how Nepal’s news coverage of the COVID-19 pandemic contributed to the creation of a moral panic targeting marginalised groups, branding them a threat as carriers of COVID-19. Such media representations instilled fear and feelings of social isolation among vulnerable populations, consequently women, Dalits, the poor and less educated migrants suffered more. Governments and other agencies must understand the patterns of COVID-19–related rumours circulating in the community so that appropriate health promotion and risk communication messages can be implemented. Federal and provincial governments and mainstream Nepalese media should dispel the current infodemic of rumours and mis/disinformation of social media regarding migrant workers and Muslims as virus carriers and spreaders. A strong government resilience-building and surveillance system should be promoted to track rumours or misinformation associated with COVID-19 in order to minimise media-induced moral panic and the othering practices faced by returnee migrants from India and the Muslim community during the COVID-19 pandemic.

## Figures and Tables

**Table 1 ijerph-19-08986-t001:** Characteristics of the interview participants (N = 30).

Characteristics	N (%)
**Gender**	
*Male*	25 (83.3)
*Female*	5 (16.7)
**Mean age**	31.1 years (Range 18–65 years)
**Caste/ethnicity**	
*Dalit*	2 (6.7)
*Janajati*	1 (3.3)
*Muslim*	17 (56.7)
*Madeshi*	10 (33.3)
**Education**	
*Illiterate*	4 (13.3)
*Literate (able to read and write)*	2 (6.7)
*Primary (Grade 1–5)*	4 (13.3)
*Secondary education (Grade 6–10)*	13 (43.3)
*Higher secondary and above (Grade 11+)*	7 (23.3)
**Working country**	
**Nepal (district)**	
*Banke/Kapilbastu/Chitwan*	12 (40)
**India (states)**	
*Kerala, Maharashtra, Gujarat, Delhi, Utter Pradesh, Panjab*	18 (60)

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
