# Peer review of "Fear, Stigma and Othering: The Impact of COVID-19 Rumours on Returnee Migrants and Muslim Populations of Nepal"

_ijerph, 2022, doi:10.3390/ijerph19158986_

Round 1
Reviewer 1 Report
The paper is quite social relevant and interesting, reflecting the experience of discrimination and resilience facing covid 19. Some minor adjustments may improve even the overall quality of the manuscript. Even though the framework of moral panics, stigmatization and othering are quite interesting and fit the goals of the study, In the literature review, it would be important to deepen some constructs and mature some arguments. First. no definition of resilience is provided, and according to the results and discussion, the authors seem to adopt a trait innate conception of resilience, assuming that participants did not seem resilient in their coping strategies or demonstrate "poor" resilience; it would be advisable to at least consider that it was an immediate response to an unprecedented pandemic situation when resources were not available, and most of the previous support networks contributed, even more, to risk through their stigmatization, as the authors highlight. Probably the strategies adopted were the best response possible, considering the situation, and simply stated authors seem to blame the victims for their own lack of resilience. Second, an increasing number of studies focus on hate speech when referring to discrimination and stigma on social media, including its increase during covid 19: authors may consider at least having a reference to that literature field. Third, gender is slightly mentioned during the manuscript, and it would be essential to have a reflection on that in the discussion/limitations section, in particular attending to their under-representation in the sample, probably associated with their prescribed role within the Muslim religion. Forth and finally, considering moral panic and stigmatization, it is not the first time that the Muslim community are perceived as a threat: the analogy to terrorism is clear in the discussion when authors mention corona terrorism, but in order to underline the implications of this study beyond the covid 19 pandemic, it would be interesting also to cite some previous literature regarding moral panics, stigma, and/or discrimination regarding Muslim community, namely associated with terrorism.
Author Response
The paper is quite social relevant and interesting, reflecting the experience of discrimination and resilience facing covid 19. Some minor adjustments may improve even the overall quality of the manuscript. Even though the framework of moral panics, stigmatization and othering are quite interesting and fit the goals of the study, In the literature review, it would be important to deepen some constructs and mature some arguments. First. no definition of resilience is provided, and according to the results and discussion, the authors seem to adopt a trait innate conception of resilience, assuming that participants did not seem resilient in their coping strategies or demonstrate "poor" resilience; it would be advisable to at least consider that it was an immediate response to an unprecedented pandemic situation when resources were not available, and most of the previous support networks contributed, even more, to risk through their stigmatization, as the authors highlight. Probably the strategies adopted were the best response possible, considering the situation, and simply stated authors seem to blame the victims for their own lack of resilience.
Authors’ response: We have deepened our arguments re. moral panics in the Introduction (see page 2; line 47-49; 54-56; 64-69; 74-75 ), we agree that resilience was not main issue and we have removed it from the text, to remove any idea that we might be victim-blaming. We also added the positive statement: “Our findings show both migrants and Muslims felt forced to take safeguarding actions, i.e. doing the best they could under the circumstances. Such actions included self-imposed home quarantine, crossing the border through informal routes” (see line 488…)
Second, an increasing number of studies focus on hate speech when referring to discrimination and stigma on social media, including its increase during covid 19: authors may consider at least having a reference to that literature field.
Authors’ response: Thank you for your suggestion. We have added references (line 475)
Third, gender is slightly mentioned during the manuscript, and it would be essential to have a reflection on that in the discussion/limitations section, in particular attending to their under-representation in the sample, probably associated with their prescribed role within the Muslim religion.
Authors’ response : We have acknowledge this on page 13 ( see line 548-549).
Forth and finally, considering moral panic and stigmatization, it is not the first time that the Muslim community are perceived as a threat: the analogy to terrorism is clear in the discussion when authors mention corona terrorism, but in order to underline the implications of this study beyond the covid 19 pandemic, it would be interesting also to cite some previous literature regarding moral panics, stigma, and/or discrimination regarding Muslim community, namely associated with terrorism.
Authors’ response: We have added references and text related to Muslims being lined to terrorism. We have made links to moral panics related to AIDS, Ebola and SARS (see line 86, 64,65)
Reviewer 2 Report
The methodology is well described and justified but theory could be improved. A deeper discussion about moral panic, inclusively, giving examples historical and geographically diverse, would be wanted. I also suggest the following reference:
KLEINMAN, A. (1988). The Illness Narratives: Suffering, Healing, And The Human Condition. New York: Basic Books. ISBN 978-0465032044
The authors should review the text in terms of language. Among other things, I found repeated words that synonyms could replace (Ex. in lines 518 and 520 the word "However" appears twice at the beginning of the sentences).
Finally, the references must be reviewed and corrected. It doesn't follow an alphabetic order?
Author Response
The methodology is well described and justified but theory could be improved. A deeper discussion about moral panic, inclusively, giving examples historical and geographically diverse, would be wanted. I also suggest the following reference:
KLEINMAN, A. (1988). The Illness Narratives: Suffering, Healing, And The Human Condition. New York: Basic Books. ISBN 978-0465032044
Authors’ response: We have deepened discussion around moral panics and we have woven the Kleinman reference in the resubmitted text (see page 2; line 47-49; 54-56; 64-69; 74-75 ). As suggested, Kleinman reference has been added (see line 61)
The authors should review the text in terms of language. Among other things, I found repeated words that synonyms could replace (Ex. in lines 518 and 520 the word "However" appears twice at the beginning of the sentences).
Authors’ response: Thank you. Proof-reading of the manuscript has been done.
Finally, the references must be reviewed and corrected. It doesn't follow an alphabetic order?
Authors’ response: Thank you. We have followed the journal’s instructions.
Reviewer 3 Report
The manuscript titled "Fear, Stigma and Othering: The Impact of COVID-19 Rumours on Returnee Migrants and Muslim Population of Nepal" captures the essence of the variables indicated in the title. The findings are articulate and plausible. However, I have made some minor suggestions in the manuscript for authors to consider.

Author Response
The manuscript titled "Fear, Stigma and Othering: The Impact of COVID-19 Rumours on Returnee Migrants and Muslim Population of Nepal" captures the essence of the variables indicated in the title. The findings are articulate and plausible. However, I have made some minor suggestions in the manuscript for authors to consider.
Authors’ response: Thank you for your kind comments, all minor suggestions have been included.
Reviewer 4 Report
The topic of the paper is important, but the current paper organization makes the academic contribution of the paper ambiguous. Please check the following comments;
1. line 45 to 57 or the beginning of chapter 2; showing the difference of the current work with references and the critics to them
The development of "moral panic" is somewhat summarized, while they gives too abstract to clarify the novelty or contribution of this research. You should add more references and in-depth comments about the existing studies, otherwise the academic contribution of this paper is quite unclear.
2. line 88 to 100; explanations about the anticipated questions or topics of interviewees
In case of qualitative study by interview, the detailed design of questions wold be difficult, but some of critical topics in the interview should be presented, otherwise the aim of researcher and the main concern of the current study becomes ambiguous. In this study, only about the attributes of interviewers and interviewees were shown while the anticipated questions are never be shown prior to the result of studies. This is definitely out of style of an academic paper organization.
3. Section 3 & 4; missing about the comparative discussion with existing studies and novel findings of current work
The style of discussion is just like "following-up" the interviewed contents, which are not stressed about the similarity / difference of findings between the previous studies and the obtained interviews. In order to clarify the novelty of current study, it is necessary to show some critical points of concern as the "research questions" before starting the discussion section, which are obtained through the critics to the existing studies. The poor organization of current chapters harms the value of this work. As a result, the discussion or conclusion of this paper seems "weak", as an academic paper.
Author Response
The topic of the paper is important, but the current paper organization makes the academic contribution of the paper ambiguous. Please check the following comments;
1. line 45 to 57 or the beginning of chapter 2; showing the difference of the current work with references and the critics to them
The development of "moral panic" is somewhat summarized, while they gives too abstract to clarify the novelty or contribution of this research. You should add more references and in-depth comments about the existing studies, otherwise the academic contribution of this paper is quite unclear.
Authors’ response: Thank you, three of the four reviewers asked for more detailed discussion of the concepts, we have expanded and deepened the arguments (see page 2; line 47-49; 54-56; 64-69; 74-75, 86)
2. line 88 to 100; explanations about the anticipated questions or topics of interviewees
In case of qualitative study by interview, the detailed design of questions would be difficult, but some of critical topics in the interview should be presented, otherwise the aim of researcher and the main concern of the current study becomes ambiguous. In this study, only about the attributes of interviewers and interviewees were shown while the anticipated questions are never be shown prior to the result of studies. This is definitely out of style of an academic paper organization.
Authors’ response: We have added that the interview schedules are available from the first author, and we have summarised the key topics addressed in the interviews (line 120-123).
3. Section 3 & 4; missing about the comparative discussion with existing studies and novel findings of current work
The style of discussion is just like "following-up" the interviewed contents, which are not stressed about the similarity / difference of findings between the previous studies and the obtained interviews. In order to clarify the novelty of current study, it is necessary to show some critical points of concern as the "research questions" before starting the discussion section, which are obtained through the critics to the existing studies. The poor organization of current chapters harms the value of this work. As a result, the discussion or conclusion of this paper seems "weak", as an academic paper.
Authors’ response: We have added more literature to make this important comparison, thank you. For example, please line starting 457, 472, 486, 492, 497, 516.
Round 2
Reviewer 4 Report
Paper organization has been substantially improved. The discussion in conclusion, answering to the raised significance of this study in line 273 to 274, is still weak, however.
Author Response
Thank you for your suggestion. We have now included two sentences (discussion and conclusion) to highlight that Dalits, female, the poor, and less educated migrants suffered more.